# Mac-/Lactosylceramide regulates intestinal homeostasis and secretory cell fate commitment by facilitating Notch signaling

Kebei Tang[1,2†], Xuewen Li[1†], Jiulong Hu[1,3,4†], Jingyuan Shi[1], Yumei Li[1], Yansu Chen[1], Chang Yin[1], Fengchao Wang[1,5], Rongwen Xi[1,5*]

[1]National Institute of Biological Sciences, Beijing, China; [2]Academy for Advanced Interdisciplinary Studies, Peking University, Beijing, China; [3]School of Life Sciences, Tsinghua University, Beijing, China; [4]School of Basic Medicine, Nanchang Medical College, Nanchang, China; [5]Tsinghua Institute of Multidisciplinary Biomedical Research, Tsinghua University, Beijing, China

*For correspondence: xirongwen@nibs.ac.cn

†These authors contributed equally to this work.

Competing interest: The authors declare that no competing interests exist.

## eLife Assessment

This **important** study provides **convincing** evidence that glucosylceramide synthase (GlcT), a rate-limiting enzyme for glycosphingolipid (GSL) production, plays a role in the differentiation of intestinal cells. Mutations in GlcT compromise Notch signaling in the *Drosophila* intestinal stem cell lineage, resulting in the formation of enteroendocrine tumors. Further data suggest that a homolog of glucosylceramide synthase also influences Notch signaling in the mammalian intestine. While the outstanding strengths of the initial genetic and downstream pathway analyses are noted, there are minor weaknesses in the data regarding the potential role of this pathway in Delta trafficking. Nevertheless, this study opens the way for future mechanistic studies addressing how specific lipids modulate Notch signaling activity.

**Abstract** Cell-to-cell communication via Delta-Notch signaling is widely used in various tissues and organs to regulate development and patterning; however, the mechanisms regulating Notch signaling for precise cell fate decisions remain poorly understood. Similar to mammals, the intestinal stem cells in the adult *Drosophila* midgut generate both absorptive and secretory cell progeny, guided by differential levels of Notch activation. Here we performed a forward genetic screen in *Drosophila* and identified glucosylceramide synthase (GlcT), a rate-limiting enzyme for glycosphingolipid (GSL) production, whose mutation causes the development of secretory cell tumors. Genetic analysis of the GSL synthesis pathway, combined with metabolite rescue experiments, revealed that the tumor formation is linked to a deficiency in Mactosylceramide/Lactosylceramide. This deficiency impaired the endocytic recycling of the Delta, subsequently reducing Notch signaling activation. Conditional knockout of *Ugcg*, the mammalian ortholog of *GlcT*, in mouse small intestine caused an excessive differentiation of goblet cells, phenotypes similar to these caused by Notch inhibition. Our study suggests an evolutionarily conserved role for a specific GSL metabolite in modulating Notch signaling during stem cell fate decisions and provides a molecular connection between ceramide metabolism and Notch signaling in regulating tissue homeostasis and tumor formation.

## Introduction

Notch signaling is an evolutionarily conserved pathway in metazoans that mediates local cell–cell interactions through the membrane-tethered ligand Delta (Dl) and the membrane receptor Notch. This pathway transduces signals from the cell surface to the nucleus, regulating the transcription of target genes (*Bray, 2006*; *Kopan and Ilagan, 2009*). Dl-Notch signaling controls various cell fates and developmental processes, with mutations in the pathway implicated in numerous human diseases, including congenital defects and cancer (*Artavanis-Tsakonas et al., 1999*; *Penton et al., 2012*; *Bolós et al., 2007*). Given that Notch can transduce, amplify, and consolidate molecular differences to influence cell fate decisions, it is crucial to understand how the strength of the Notch signaling pathway is precisely regulated to ensure proper cell fate determination and tissue development.

Notch signaling is known as a major regulator of cell fate decisions in both mammalian and *Drosophila* intestinal stem cell (ISC) lineages, as its activity determines the binary fate of intestinal progenitor cells. A high level of Notch activity promotes the specification of absorptive enterocytes, while low or absent Notch activity favors the specification of secretory cell types (*Boumard and Bardin, 2021*; *Koch et al., 2013*). The *Drosophila* ISC lineage in the adult midgut is relatively simple, comprising only one type of secretory cell: enteroendocrine cells (EEs). This simplicity makes it an attractive experimental system for studying Notch signaling in cell fate decisions during epithelial renewal (*Boumard and Bardin, 2021*). The fly ISCs, which specifically express Dl, periodically generate two types of committed daughter cells: enteroblasts, which experience high levels of Notch activation and are primed for the specification of absorptive enterocytes, and enteroendocrine progenitor cells (EEPs), which receive low levels of Notch activation. Typically, EEPs undergo one round of mitosis before terminal differentiation, resulting in the formation of EE pairs (*Ohlstein and Spradling, 2007*; *Ohlstein and Spradling, 2006*; *Chen et al., 2018*; *Micchelli and Perrimon, 2006*). The loss of Notch signaling results in a complete blockage of enteroblast differentiation, leading to the accumulation of ISCs, EEPs, and EEs in the intestinal epithelium, producing an 'EE tumor' phenotype (*Ohlstein and Spradling, 2007*; *Chen et al., 2018*). It is suggested that the expression level of Dl in individual ISCs varies, leading to differential levels of Notch activation in the immediate daughter cells (*Ohlstein and Spradling, 2006*). This variation in Dl expression may result from dynamic bidirectional Dl-Notch signaling between ISCs and EEPs, which in parallel or in conjunction with an intrinsic feedback mechanism in ISCs, guides a periodic activation of an EE fate inducer Scute, thereby periodic generation of EEPs from ISCs (*Chen et al., 2018*; *Guo and Ohlstein, 2015*; *Henrique and Schweisguth, 2019*). The transient activation of Scute induces irreversible Prospero expression in EEPs through a Phyl-Sina-Ttk69 regulatory cascade, initiating terminal differentiation into EEs (*Chen et al., 2018*; *Yin and Xi, 2018*; *Wang et al., 2015*; *Guo et al., 2022*). During the period of Scute inactivation, ISCs only generate enterocyte-committed enteroblasts by default via Dl-Notch mediated lateral inhibition (*Ohlstein and Spradling, 2007*; *de Navascués et al., 2012*). In newly enclosed flies, increased lipid intake from food can alter the membrane trafficking of Dl, subsequently increasing the frequency of EE specification from ISCs (*Obniski et al., 2018*). Thus, both intrinsic and extrinsic mechanisms are involved in regulating Dl-Notch signaling to orchestrate stem cell fate decisions.

To better understand fate regulation in the *Drosophila* ISC lineage, here we conducted a forward mosaic genetic screen in the adult *Drosophila* midgut, focusing on the right arm of the second chromosome to identify genes whose mutations could disrupt ISC fate decisions. From this screen, we identified several complementation groups that result in EE tumor phenotypes. While most of the identified gene loci correspond to known components of the Notch signaling pathway, we discovered one gene locus not previously associated with Notch signaling. Further analyses revealed a specific GSL metabolite that regulates Notch signaling and cell fate decisions in the intestinal epithelium, and this function appears to be conserved from *Drosophila* to mice.

## Results

### A mosaic genetic screen identifies *GlcT* as a tumor suppressor in the adult *Drosophila* midgut

We performed EMS mutagenesis on chromosome 2R (carrying FRTG13/42B) and used the Mosaic Analysis with a Repressible Cell Marker (MARCM) system to induce mutant clones in the adult midgut (*Figure 1—figure supplement 1*). We screened approximately 10,000 lethal lines and identified a

total of 12 mutations across six complementation groups that exhibited a 'small cell tumor' pheno-type, characterized by a significant accumulation of diploid cells within the clones (*Figure 1—figure supplement 1* and *Supplementary file 1*). Co-staining with the ISC marker Dl and the EE cell marker Pros revealed that most of these tumors contained an excessive number of ISCs and EEs, the EE tumor phenotype commonly observed in Notch mutant clones (*Micchelli and Perrimon, 2006*; *Perdigoto et al., 2011*). Further genetic mapping showed that three complementation groups were associated with genes known to function in the Notch pathway: *mam*, *Gmer*, and *O-fut1* (*Ayukawa et al., 2012*; *Petcherski and Kimble, 2000*; *Sasamura et al., 2007*). One complementation group, comprising two alleles (*EA30* and *E230*), mapped to the *GlcT* gene (see details in 'Materials and methods'), while the remaining two complementation groups, each consisting of one allele, remain undetermined (*Supplementary file 1*). *GlcT* encodes glucosylceramide synthase, an enzyme that catalyzes the formation of glucosylceramide, the core component of glycosphingolipids (GSLs). Both alleles carried one or more missense mutations resulting in the amino acid replacements E292K (for *EA30*) and L395P/S397T (for *E230*). The two mutations produced a similar tumor phenotype in terms of tumor size and the compo-sition of EEs in the tumors (*Figure 1B–D*). The large polyploid cells were still present in the tumors, indicating that the multipotency of ISCs was retained, but the differentiation of ISCs may have been biased towards the EE fate commitment. Additionally, $GlcT^{\Delta 8}$, a previously characterized null allele of *GlcT* (*Satoh et al., 2013*), exhibited a similar EE tumor phenotype (*Figure 1B–D*). Expressing a UAS-GlcT transgene in the mutant clones fully prevented the tumor phenotype (*Figure 1E and F*). Further-more, depleting *GlcT* in progenitor cells using RNAi with $esg\text{-}GAL4^{ts}$ also led to overproliferation of ISCs and excessive generation of EEs in the midgut epithelium (*Figure 1G–J*). Taken together, these data suggest that *GlcT* acts as a tumor suppressor in adult *Drosophila* midgut. Loss of *GlcT* induces ISC over-proliferation and excessive EE differentiation, resulting in an EE tumor or 'neuroendocrine tumor (NET)' phenotype.

## Genetic analysis of the GSL synthesis pathway components reveals specific tumor-suppressive activity for *egh*, in addition to *GlcT*

GlcT is involved in the initial step in the GSL biosynthesis pathway, which synthesizes glucosylcer-amide by transferring glucose to ceramide (*Figure 2A*). The loss of *GlcT* could result in the accu-mulation of ceramide or the absence of glucosylceramide and the downstream GSL metabolites. Ceramide accumulation has been linked to promoting cell apoptosis (*Selzner et al., 2001*), which could potentially explain the ISC over-proliferation phenotype since apoptotic cells can induce compensatory ISC proliferation (*Ryoo et al., 2004*). However, we did not observe a significant increase in apoptotic cells within the mutant clones (*Figure 2B*). Moreover, the overexpression of p35, an inhibitor of cell apoptosis, failed to suppress or alleviate the tumor phenotype (*Figure 2B*). Ceramidase (CDase) catalyzes ceramide into sphingosine and free fatty acid, and overexpressing *CDase* can in theory downregulate ceramide levels. We therefore introduced *CDase* expression in the mutant clones, but this intervention also did not alleviate the tumor phenotype (*Figure 2B*). Hence, the EE tumors developed from *GlcT* mutant clones are unlikely caused by the accumulation of ceramide.

In the GSL biosynthesis pathway in *Drosophila*, glucosylceramide undergoes sequential modifi-cations catalyzed by Egghead (Egh) and Brainiac (Brn), which add the second (Mactosyl-) and third (GlcNAc-) glycosyl residues, respectively (*Figure 2A*; *Wandall et al., 2005*; *Müller et al., 2002*). This leads to the synthesis of Mactosylceramide (MacCer) and GlcNAcβ1-3Manβ1-4Glcβ1Ceramide. Complex GSL biosynthesis involves the participation of β4-N-acetylgalactosyltransferase (β4Gal-NAcTA/TB) and α1,4-galactosyltransferase (α4GT1/2) in the fourth and fifth steps of GSL sugar chain elongation (*Stolz et al., 2008*; *Chen et al., 2007*). These enzymes catalyze the formation of GalNAcβ1-4GlcNAcβ1-3Manβ1-4Glcβ1Ceramide and GalNAc-α1-4GalNAcβ1-4GlcNAcβ1-3Manβ1-4Glcβ1Ceramide, respectively.

To understand the consequences of mutations in these GSL biosynthesis pathway genes, we gener-ated mutant MARCM clones for each of the above-mentioned genes and examined their effects. Interestingly, mutations in *egh*, assessed using three different loss-of-function alleles, consistently resulted in cell over-proliferation and excessive EE phenotypes, similar to those caused by the loss of *GlcT* (*Figure 2E–G*). In contrast, none of the mutant clones of *brn*, *β4GalNAcTA*, or *α4GT1* exhibited any noticeable abnormalities in clone size or percentages of EEs, when compared to the control group

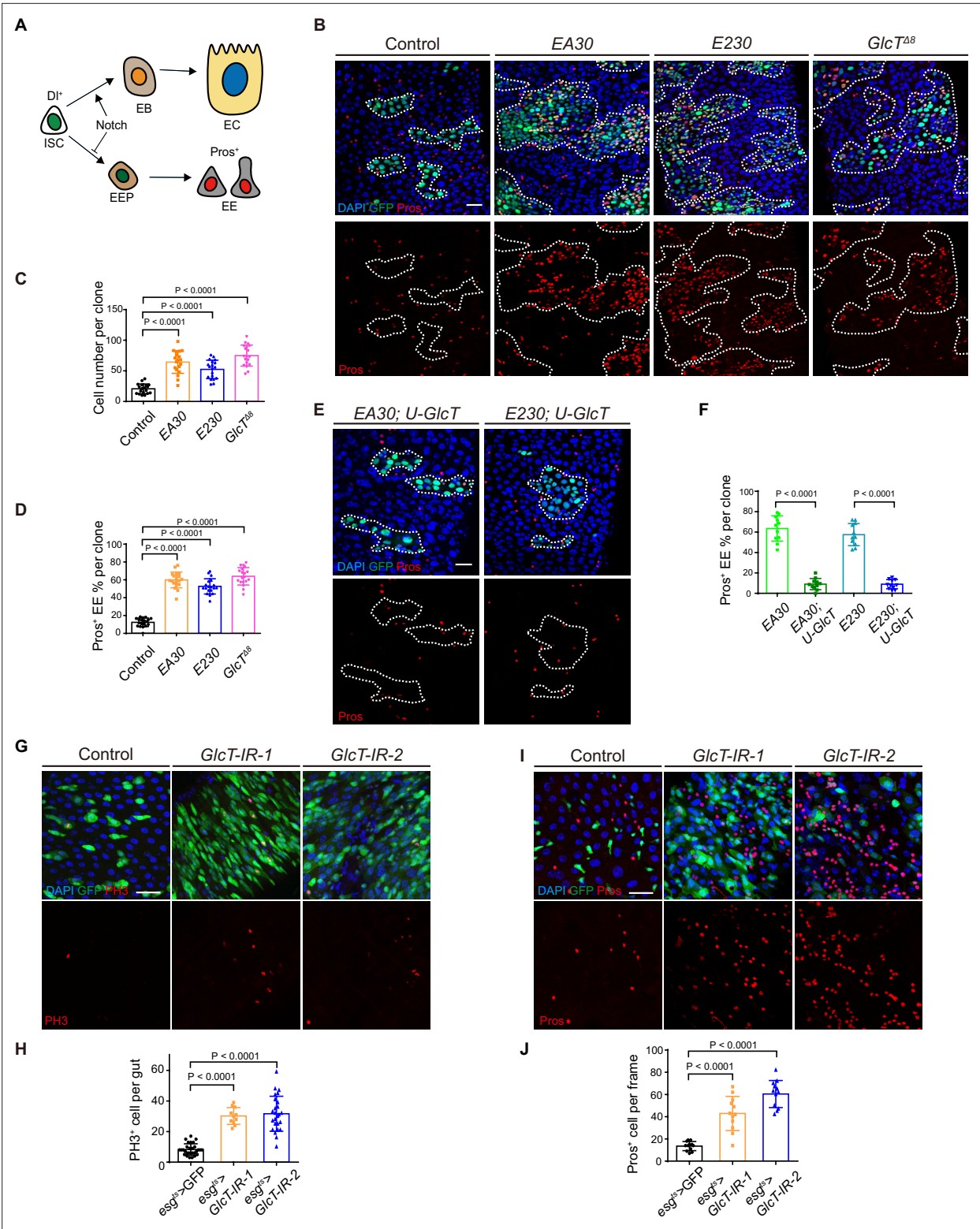

**Figure 1.** A mosaic genetic screen identifies *GlcT* as a tumor suppressor in adult *Drosophila* midgut. (**A**) Diagram showing the lineage hierarchy of intestinal stem cells (ISCs) in the *Drosophila* intestine. EB, enteroblast; EEP, enteroendocrine progenitor; EC, enterocyte; EE, enteroendocrine cell. (**B–D**) Mosaic Analysis with a Repressible Cell Marker (MARCM) clones (green) induced for 5 days with Pros (red) staining (**B**), together with quantification of cell number (**C**, n=6) and the percentage of Pros+ EE cells per clone (**D**, n=6) in *EA30*, *E230*, and *GlcT* null allele (*GlcT^Δ8*) mutant. (**E, F**) Overexpression of *GlcT* in *EA30* or *E230* mutant clones with Pros staining, along with quantification of the percentage of Pros+ EE cells per clone (**F**, n=4). (**G–J**) *esg*

*Figure 1 continued on next page*

Figure 1 continued

>*GlcT* IR in adult fly gut for 14 days. PH3 staining (**G**) and Pros staining (**I**) with quantification of positively stained cells (**H**, n=11; **J**, n=12). Error bars represent mean ± SEM, with p-values indicated in the figure (two-tailed Student's *t*-test). Scale bars: 25 μm.

The online version of this article includes the following figure supplement(s) for figure 1:

**Figure supplement 1.** Cross-scheme for the mosaic genetic screen on the 2R chromosome.

of wild-type clones (*Figure 2E–G*). Therefore, GlcT and Egh, but not other enzymes in the GSL biosynthesis pathway, exhibit specific tumor suppressive activity in the adult *Drosophila* midgut.

## The EEC tumor phenotype is a result of MacCer deficiency

As Egh is necessary for the biosynthesis of MacCer, the specific phenotype resulted from the loss of GlcT and Egh but not Brn could be attributed to the absence of MacCer. To test this hypothesis, we fed flies carrying *GlcT* mutant clones with Lactosylceramide (LacCer), an analog of MacCer that functions similarly to MacCer in mammalian cells. Remarkably, LacCer significantly suppressed the overgrowth and excessive EE phenotype in the anterior midgut clones (*Figure 3*). The suppression effect was also significant, though relatively mild, in the posterior midgut (*Figure 3*). This difference may be attributed to substantial absorption or metabolism of LacCer prior to its entry into the posterior midgut. Therefore, among the various GSL metabolites, MacCer and LacCer appear to possess specific tumor-suppressive activities in the *Drosophila* midgut.

## Loss of *GlcT* leads to reduced activation of Notch signaling in ISC progenies

In adult *Drosophila* midgut, the proliferation and differentiation of ISCs are regulated by multiple signaling pathways. Among these, the EGFR/Ras/MAPK and JAK/STAT signaling pathways play significant roles in promoting ISC proliferation during intestinal renewal and regeneration following tissue damage or infection (*Xu et al., 2011*; *Jiang et al., 2009*). Additionally, Notch signaling not only regulates ISC proliferation but also controls fate decisions between EE/EC during ISC differentiation (*Perdigoto et al., 2011*; *Ohlstein and Spradling, 2007*; *Guo and Ohlstein, 2015*). Loss of Notch leads to increased ISC proliferation and excessive EE generation, a phenotype similar to those caused by *GlcT* mutations. Therefore, we examined whether the loss of *GlcT* affects the activities of these signaling pathways.

Normally, among the intestinal cells in the midgut epithelium, progenitor cells including ISCs and enteroblasts exhibit low but specific activation of EGFR and JAK/STAT activities (*Xu et al., 2011*; *Jiang et al., 2009*; *Jiang et al., 2011*). The growth of *GlcT* mutant clones appeared to cause a general increase of EGFR and JAK/STAT activities within the intestinal epithelium, as many differentiated cells also exhibited EGFR or JAK/STAT signaling activation (*Figure 4—figure supplement 1*), which is likely due to stress responses induced by tumor growth (*Patel et al., 2015*). Nevertheless, we observed that pERK, a direct indicator of EGFR/Ras/MAPK signaling activity, was not significantly increased in *GlcT* mutant cells compared to cells outside the clones (*Figure 4—figure supplement 1*). Similarly, the nuclear expression of STAT92E, which reflects JAK/STAT pathway activation, did not show significant changes in *GlcT* mutant cells compared to normal cells outside the clones (*Figure 4—figure supplement 1*). However, the expression of *NRE-lacZ*, a reporter for Notch pathway activation, was specifically down-regulated in *GlcT* mutant clones. Most cells within the clones exhibited lower LacZ expression levels compared to *NRE-lacZ* positive cells outside the clones (*Figure 4A and B*). Previous studies have shown that while the loss of function of Notch completely blocks EC differentiation from ISCs, a reduction in Notch activity does not necessarily hinder EC differentiation from ISCs. Instead, it may lead to a dose-dependent increase in the ratio of differentiated EEs to ECs, depending on the severity of Notch loss (*Perdigoto et al., 2011*). Therefore, the increased EE to EC ratio and the reduced NRE-lacZ expression observed in *GlcT* mutant clones collectively suggest that the mutant phenotype may be caused by a reduction in Notch signaling activity.

In the ISC lineage, Notch signaling not only guides EC/EE fate decisions, but is also participated in regulating EE subtype diversity. Typically, each EEP undergoes one round of asymmetric cell division to yield a pair of distinctive EEs, an allatostatin C (AstC)[+] class I EE and a tachykinin (Tk)[+] class II EE (*Chen et al., 2018*). The loss of *Notch* specifically compromises Mirr-dependent specification of class

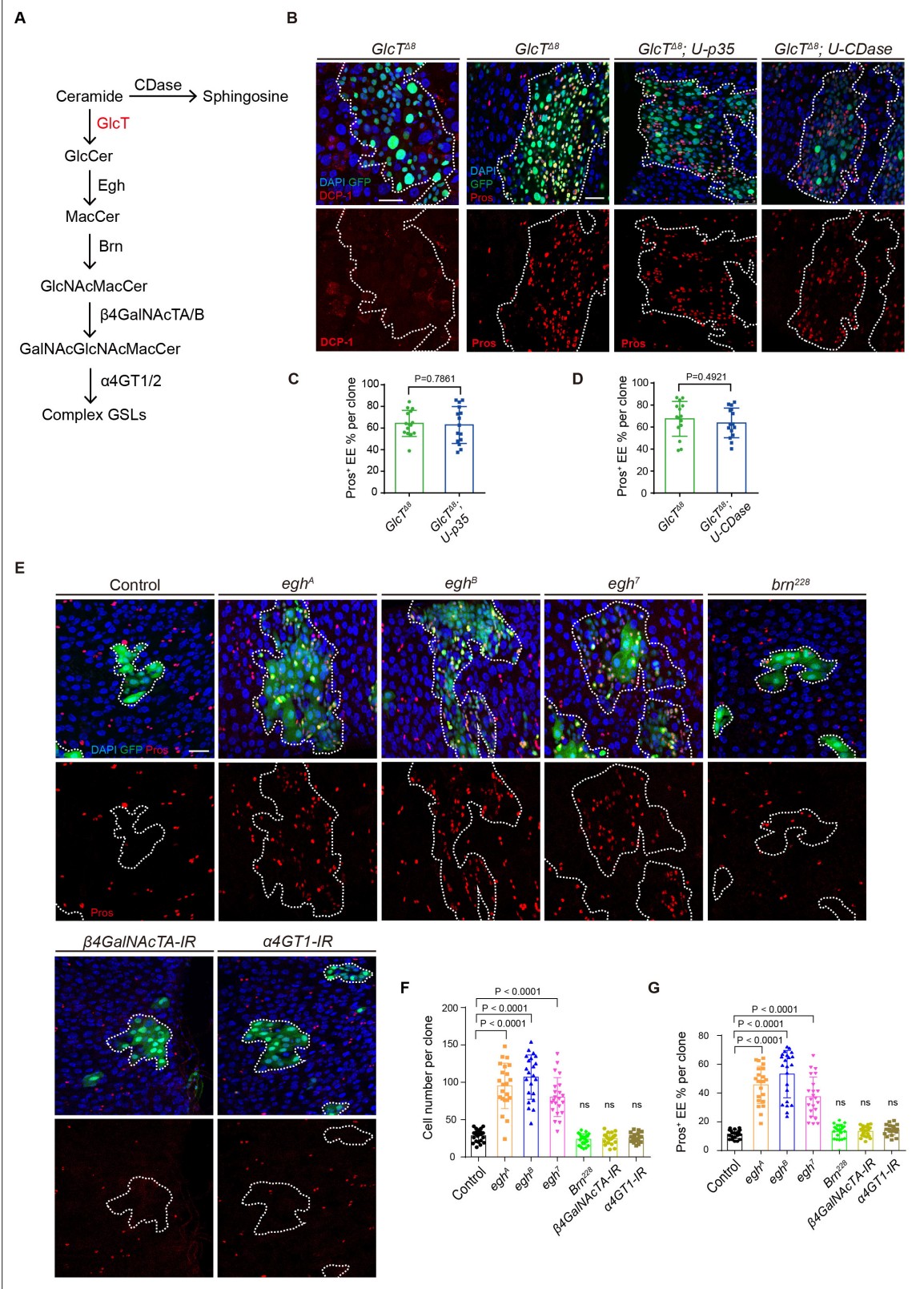

**Figure 2.** Genetic analysis of the components of the glycosphingolipid (GSL) synthesis pathway reveals specific tumor-suppressive activity for *egh*, in addition to *GlcT*. (**A**) Diagram illustrating a portion of the GSL metabolic pathway synthesized from ceramide, highlighting the key enzymes involved at each step. (**B–D**) *GlcT^Δ8^* clones induced for 7 days without or with co-expression of p35 or CDase, stained with anti-DCP-1 or anti-Pros (**B**). Quantification of the percentage of Pros⁺ cells in clones of the indicated genotypes (**C, D**, n=5). (**E–G**) Mosaic Analysis with a Repressible Cell Marker (MARCM) clones

*Figure 2 continued on next page*

*Figure 2 continued*

induced in several mutants of key enzymes in the GSL synthesis pathway. Pros staining in *egh* mutant clones (*egh^A^*, *egh^B^*, *egh^7^*), *brn^228^*, *β4GalNAcTA-IR*, and *α4GT1-IR* clones (**E**). Quantification of cell number (**F**, n=7) and the percentage of Pros+ cells (**G**, n=7) in clones of the indicated genotypes. Error bars represent mean ± SEM, with p-values indicated (two-tailed Student's *t*-test). Scale bars: 25 μm.

II EE generation, resulting in all EEs in *Notch* mutant clones becoming AstC+ class I EEs (*Guo et al., 2022*; *Beehler-Evans and Micchelli, 2015*). We found that in *GlcT* mutant clones, virtually all EEs were positive for AstC but TK (*Figure 4C*), further supporting the idea that Notch signaling is compromised in these clones.

We next tested whether forced activation of Notch in *GlcT* mutant clones could suppress the EE tumor phenotype. To achieve this, we expressed N^intra^, an active form of Notch, in *GlcT* mutant clones. As expected, this resulted in the complete suppression of cell proliferation in *GlcT* mutant clones and promoted the differentiation of these cells into ECs, leading all clones to become single-cell clones that often contained a polyploid cell (*Figure 4D*). This finding suggests that N^intra^ is epistatic to *GlcT* in the signaling transduction pathway, indicating that GlcT may regulate Notch signaling at the ligand/receptor level.

### *GlcT* regulates the endocytic trafficking of Delta

The observations above collectively suggest that full activation of Notch signaling in the ISC lineage may require a specific GSL: Mac/Lac-Cer. As transmembrane proteins, the proper internalization

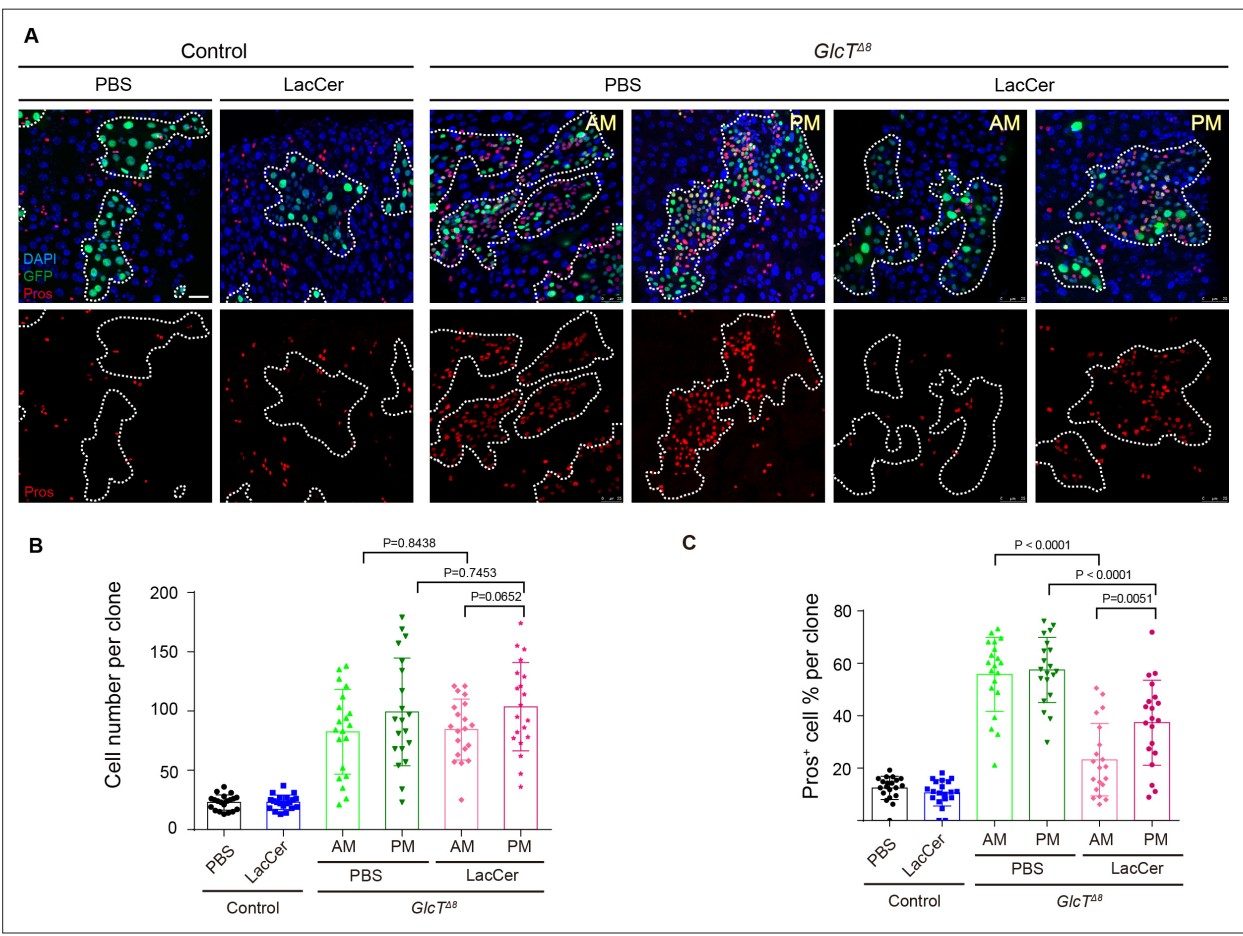

**Figure 3.** The EE tumor phenotype is a result of MacCer deficiency. (**A–C**) LacCer feeding after Mosaic Analysis with a Repressible Cell Marker (MARCM) clone induction. Anti-Pros staining of anterior midgut (AM) and posterior midgut (PM) in *GlcT^Δ8^* clones (**A**). Quantification of cell number (**B**, n=7) and the percentage of Pros+ cells (**C**, n=7) in clones of the indicated genotypes. Error bars represent mean ± SEM, with p-values indicated (two-tailed Student's *t*-test). Scale bars: 25 μm.

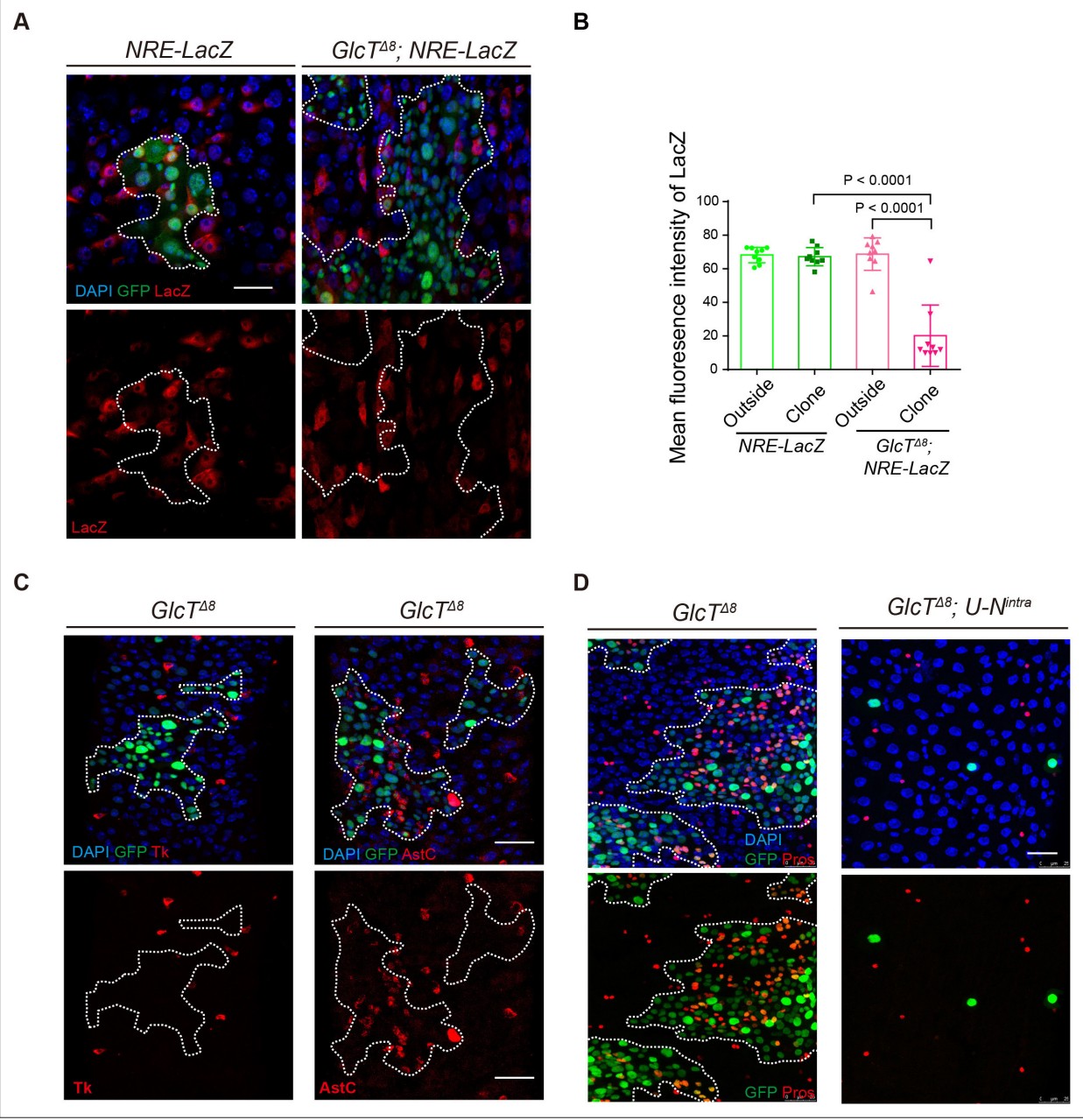

**Figure 4.** Loss of *GlcT* leads to reduced activation of Notch signaling in intestinal stem cell (ISC) progenies. (**A, B**) Expression of NRE-lacZ, the Notch activity reporter, in *GlcT^Δ8* clones (**A**) and quantification of LacZ fluorescence intensity for signal positive cell (**B**, n=3). (**C**) Tk and AstC staining in *GlcT^Δ8* clones in the posterior (R5) region of the *Drosophila* midgut. (**D**) Pros staining in *N^intra*-overexpressing *GlcT^Δ8* clones. Error bars represent mean ± SEM, with p-values indicated (two-tailed Student's *t*-test). Scale bars: 25 μm.

The online version of this article includes the following figure supplement(s) for figure 4:

**Figure supplement 1.** The EGFR/Ras/MAPK and JAK/STAT signaling activities in *GlcT* mutant clones.

and recycling of Dl and Notch are important for the proper activation of Notch signaling (***Bray and Gomez-Lamarca, 2018***; ***Yamamoto et al., 2010***). Specific lipids have been implicated in the formation of micro-domains on the cell membrane, which play a role in regulating cell–cell signaling (***Simons and Toomre, 2000***). Therefore, we asked whether the loss of MacCer caused by *GlcT* mutation could disrupt Notch signaling by affecting the stability, endocytosis, or recycling of either Dl or Notch.

Knocking down *GlcT* in ISCs using *Dl-GAL4^ts* caused an increase in EEs in the midgut epithelium (***Figure 5A and B***). However, when *GlcT* was specifically knocked down in EBs using *NRE-GAL4^ts*, there

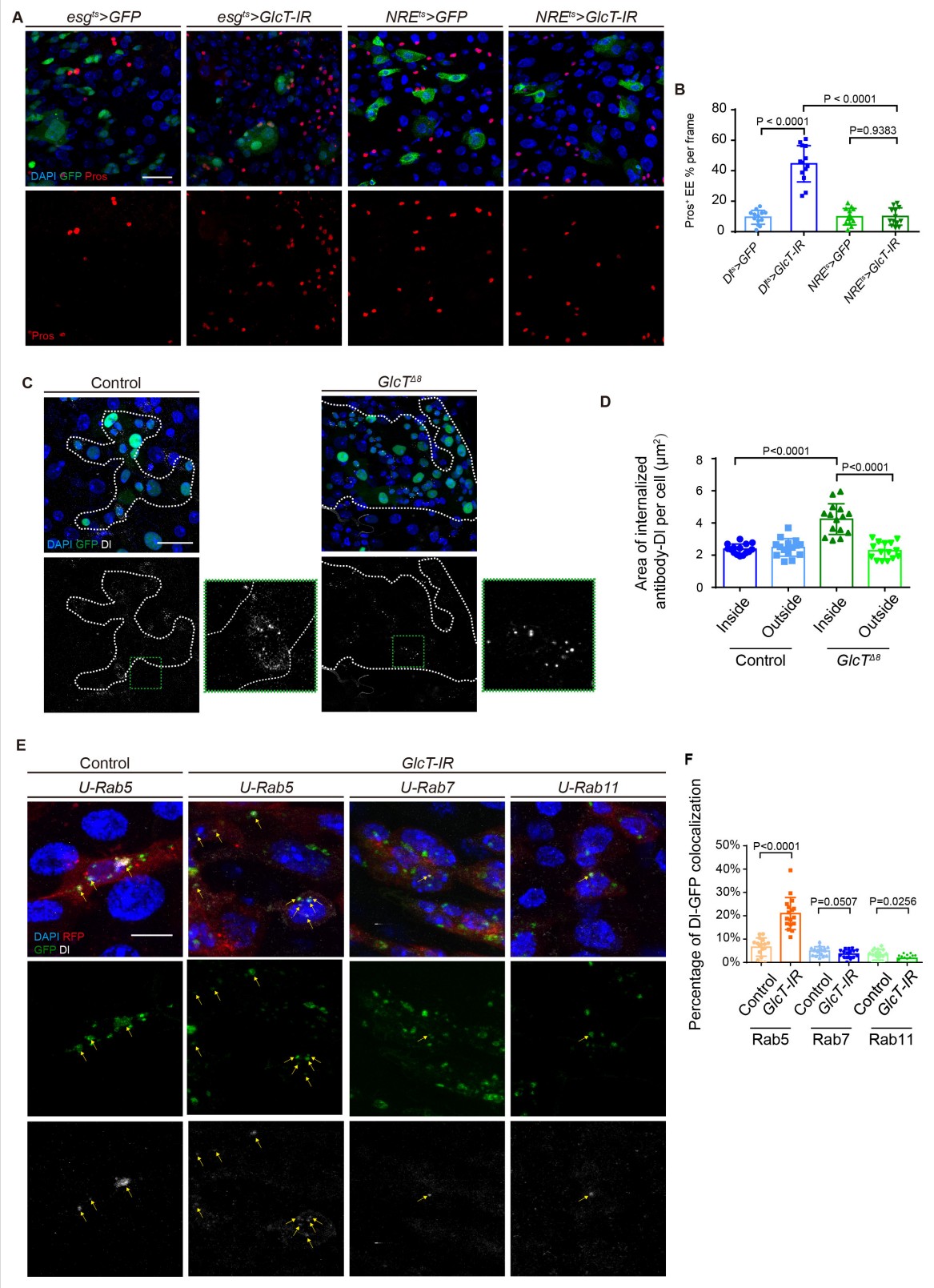

**Figure 5.** *GlcT* regulates the endocytic trafficking of Delta. (**A, B**) Pros staining in flies carrying *GlcT-IR* driven by *Dl^ts* and *NRE^ts*, respectively (**A**), and quantification of the percentage of Pros+ cells (**B**, n=4). (**C, D**) Dl staining in an antibody uptake assay performed in *GlcT^Δ8* flies after 3-hour clone induction (**C**) and quantification of internalized antibody-Dl area inside and outside of the clones (**D**, n=5). (**E, F**) Dl staining in *esg >GlcT*IR flies co-stained with Rab5-GFP, Rab7-GFP, or Rab11-GFP (**E**); yellow arrows indicate co-localization of Dl and GFP. Quantification of Dl and GFP co-localization (**F**, n=6). Error bars represent mean ± SEM, with p-values indicated (two-tailed Student's *t*-test). Scale bars: 25 μm (10 μm in **E**).

was no noticeable increase in EEs (*Figure 5A and B*). This indicates that GlcT is probably required in ISCs and implies its potential role in the regulation of Dl. However, the lack of an observable phenotype using NRE-Gal4 could be due to delayed expression, which may result in missing the critical window required for effective *GlcT* knockdown. Consequently, we cannot rule out the possibility that GlcT also plays a role in early EBs or EEPs.

It is well-known that the internalization of Dl in the signaling-sending cell is necessary for Notch activation in the signaling-receiving cell (*Bray and Gomez-Lamarca, 2018*). To investigate whether the endocytosis process of Dl is altered in *GlcT* mutant ISCs, we performed an ex vivo Dl internalization assay, following the previously described protocol (*Couturier and Schweisguth, 2014*). In this assay, we cultured and incubated the freshly dissected gut with anti-Dl antibodies, and the endocytosis of the antibody-labeled Dl was visualized after fixation and immunostaining with secondary antibodies. Interestingly, during the 3-hour period, we observed a significant difference in the subcellular localization pattern of Dl between normal and *GlcT* mutant ISCs (*Figure 5C and D*). In *GlcT* mutant ISCs, the majority of labeled Dl had already been internalized, while in wild-type ISCs, it remained on the cell membrane. This observation suggests that the loss of *GlcT* causes either reduced stability of Dl on the cell membrane or accelerated internalization of Dl from the cell membrane.

To further characterize the endocytic trafficking of Dl in *GlcT* mutant ISCs, we performed conditional depletion of *GlcT* and examined the co-localization of internalized Dl with the early endosome marker Rab5, the late endosome marker Rab7, and the recycle endosome marker Rab11, respectively. In comparison to control ISCs, *GlcT-IR* ISCs exhibited a significant increase in early endosomes containing Dl (*Figure 5E and F*). However, the percentage of late and recycle endosomes containing Dl remained similar between wild-type and *GlcT-IR* ISCs (*Figure 5E and F*). These observations suggest that the internalized Dl in *GlcT-IR* ISCs experiences a delay in early endosomes, which might cause a delay in Dl recycling and consequently a reduction in Dl-Notch signaling activity (Figure 7E).

## *GlcT* shows tissue specificity in regulating Notch signaling activity

Notch signaling has pleiotropic function and is known to be involved in the development of diverse tissue and organs. We therefore asked whether the discovered mechanism above is generally utilized in controlling Notch signaling in diverse tissues and organs. In the wing disc of third-instar larvae, Notch signaling is activated by ligands Dl and Serrate at the dorsal-ventral boundary to induce the expression of Cut (*Micchelli et al., 1997*). We found that in *GlcT* mutant clones spanning the dorsal-ventral boundary region of the wing disc, the expression of Cut was not significantly altered (*Figure 6A*). This indicates that *GlcT* is not essential for regulating Notch-mediated Cut expression in this specific context.

During oogenesis in the *Drosophila* ovary, Notch signaling regulates differentiation and transition of follicle cells from the mitotic cell cycle to the endocycle during stage 6 (*Deng et al., 2001*; *López-Schier and Johnston, 2001*). We found that in the developing egg chambers with germline mutations of *GlcT*, the patterning of the surrounding follicle cells remained largely normal (*Figure 6*). However, we observed an abnormal accumulation of Dl protein in cytoplasmic vesicles of the mutant germline cells (*Figure 6C*), indicating that there is a defect in Dl trafficking in *GlcT* mutant germline cells. However, this defect is not sufficient to impact Dl-Notch signaling to a degree where follicle cell development is adversely affected. Collectively, these observations suggest that the role for GlcT in Notch signaling does not appear to be a universal mechanism across all developmental contexts. Even in situations where this mechanism is involved, the extent to which GlcT affects Dl-Notch signaling may vary in different developmental scenarios.

## Transient loss of *Ugcg* in mouse small intestine causes reduced number of ISCs and increased number of goblet cells

As both the Notch signaling and the metabolic pathway of GSLs are conserved from *Drosophila* to mammals, we asked whether this mechanism is conserved in the mouse small intestine to regulate the proliferation and differentiation of ISCs. Notch is known to control the absorptive versus secretory cell fate of mammalian intestinal progenitors as well, and the inhibition of Notch signaling in the intestinal epithelium of mouse small intestine leads to an excessive number of goblet cells (*van Es et al., 2005*; *Riccio et al., 2008*). However, unlike in *Drosophila*, Notch positively regulates ISC self-renewal in mice. The inhibition of Notch causes ISC loss, and the excessive activation of Notch promotes

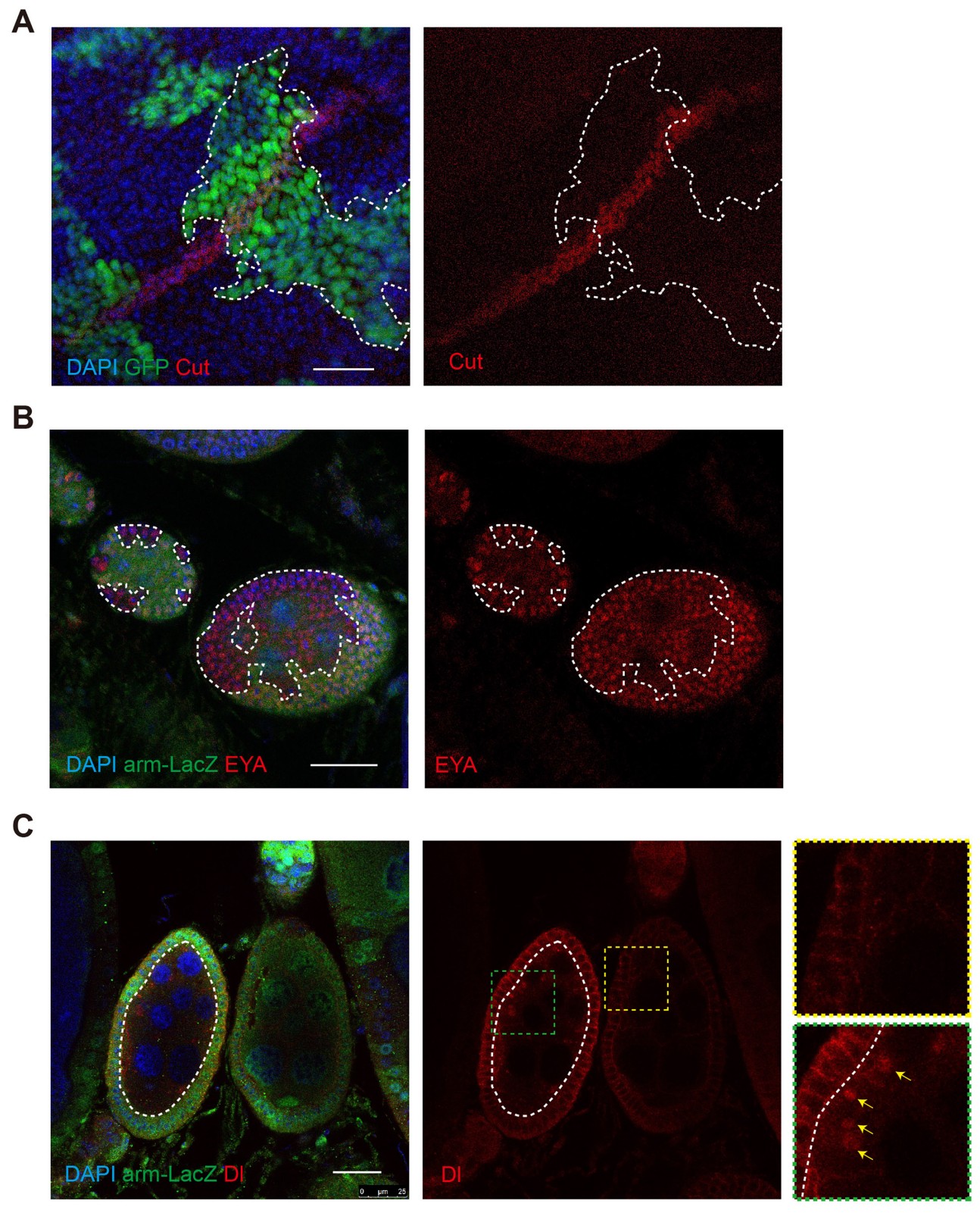

**Figure 6.** *GlcT* shows tissue specificity in regulating Notch signaling activity. (**A**) Cut staining in wing disc of *GlcT^Δ8* clones. (**B**) EYA staining in follicle cell clones (non-green areas) of *GlcT^Δ8* flies. (**C**) Dl staining in germ cell clones (non-green area) of *GlcT^Δ8* mutant clone, yellow arrows indicate punctate Dl accumulation. Scale bars: 25 μm.

the proliferation of intestinal progenitor cells (*Riccio et al., 2008*; *Fre et al., 2005*). We therefore performed conditional knockout of *Ugcg*, which encodes the mammalian glucosylceramide synthase, in the mouse small intestine and examined the consequences.

It has been previously reported that in the newborn mice with intestine-specific knockout of *Ugcg*, the proliferation of enterocytes appeared largely normal. However, 3–4 days after intestinal-specific knockout of *Ugcg* in adult mice, epithelial phenotypes including epithelial hyperproliferation, impaired lipid absorption, and the detachment of enterocytes from the basal lamina were observed, and these phenotypes are considered to be consequences of gut barrier disruption (*Jennemann et al., 2012*). To separate the primary from the secondary phenotypes, we repeated the experiment by generating *Ugcg*^flox/flox; *Villin*^CreERT2 mice and examined the phenotype at 48 hours post-tamoxifen administration, an early timepoint with a hope that the secondary phenotypes have not yet developed.

Consistent with the previous findings, the conditional knockout mice showed a severe diarrhea phenotype and died several days following tamoxifen induction. Interestingly, we observed a few phenotypes at the 48 hours timepoint that were not observed before. We found that in the mutant intestine, there was a marked increase in crypt depth accompanied by a marked reduction in villus length (*Figure 7—figure supplement 1*). In addition, an excess population of goblet cells was observed, dispersed along the crypt-villus axis (*Figure 7C and D*). By examining the expression of Olfm4, which is transcriptionally regulated by Notch in ISCs (*VanDussen et al., 2012*), we found that the fluorescence intensity of Olfm4 staining was significantly reduced in *Ugcg* mutant ISCs (*Figure 7A and B*). It is unclear how the crypt-villus morphology phenotype is developed in *Ugcg* mutant epithelium, but the increased goblet cells and the reduced expression level of Olfm4 are consistent with a Notch phenotype, suggesting a conserved role for glucosylceramide synthase in modulating Notch signaling from *Drosophila* to mammals.

## Discussion

From an unbiased genetic screen on the right arm of chromosome II, here we have pinpointed four mutant loci that cause an excessive EE phenotype in the intestinal epithelium. Three of them encode known components of the Notch signaling (*mam*, *Gmer*, and *O-fut1*), and the remaining one, *GlcT*, turns out to be a new Notch regulator as well. This work further reinforces the notion that Notch is the key regulator of the secretory versus absorptive cell fate decisions from ISCs and reveals an evolutionarily conserved link between GSL metabolism and Notch signaling in regulating intestinal homeostasis.

As *GlcT* encodes glucosylceramide synthase, which catalyzes the formation of glucosylceramide from ceramide, we further analyzed *egh* and *brn*, which encode enzymes for the subsequent sequential modifications of glucosylceramide by adding the second (Mactosyl-) and third (GlcNAc-) glycosyl residues, respectively. We found that loss of *egh*, but not *brn*, can give rise to a similar excessive EE phenotype. In addition, the phenotype caused by *GlcT* can be suppressed by feeding with LacCer, which is analogous to MacCer. Collectively, these observations demonstrate that the excessive EE phenotype caused by *GlcT* or *egh* mutation is a result of MacCer deficiency.

GSLs constitute a specific category of glycolipids localized on the outer leaflet of eukaryotic cell membranes, and studies have implicated their roles in regulating multiple cellular signaling pathways by serving on the lipid raft to facilitate signaling transductions (*Coskun et al., 2011*; *Hamel et al., 2010*; *Ideo et al., 2003*; *Park et al., 2012*). In particular, there is an evolutionarily conserved phospholipid binding domain (C2 domain) on the Notch ligand that is known to interact with GSLs, and this interaction facilitates ligand-receptor interactions and thereby promotes robust Notch signaling (*Suckling et al., 2017*; *Martins et al., 2021*; *Chillakuri et al., 2013*). Our analyses of *GlcT* mutant cells in the *Drosophila* midgut suggest that MacCer may represent a key member of GSLs in facilitating Dl-Notch signaling transduction between ISCs and their immediate progeny to steer binary cell fate decisions. To do so, MacCer may help to modulate the endocytic trafficking of Dl, an event that is known to be essential for Notch activation. We found that Dl on the membrane of *GlcT* mutant ISCs tends to be rapidly endocytosed, and the resultant endosomes appear to experience a delay in transition from early endosome to late endosome. Prior genetic investigations have linked GSLs synthesized by α4GT1 to the modulation of Dl ligand activity in vivo. While the loss of α4GT1 does not overtly affect Dl-Notch signaling, it can exacerbate the wing phenotype of haploinsufficient *Notch* mutants. Furthermore, the overexpression of α4GT1 has been shown to counteract signaling and

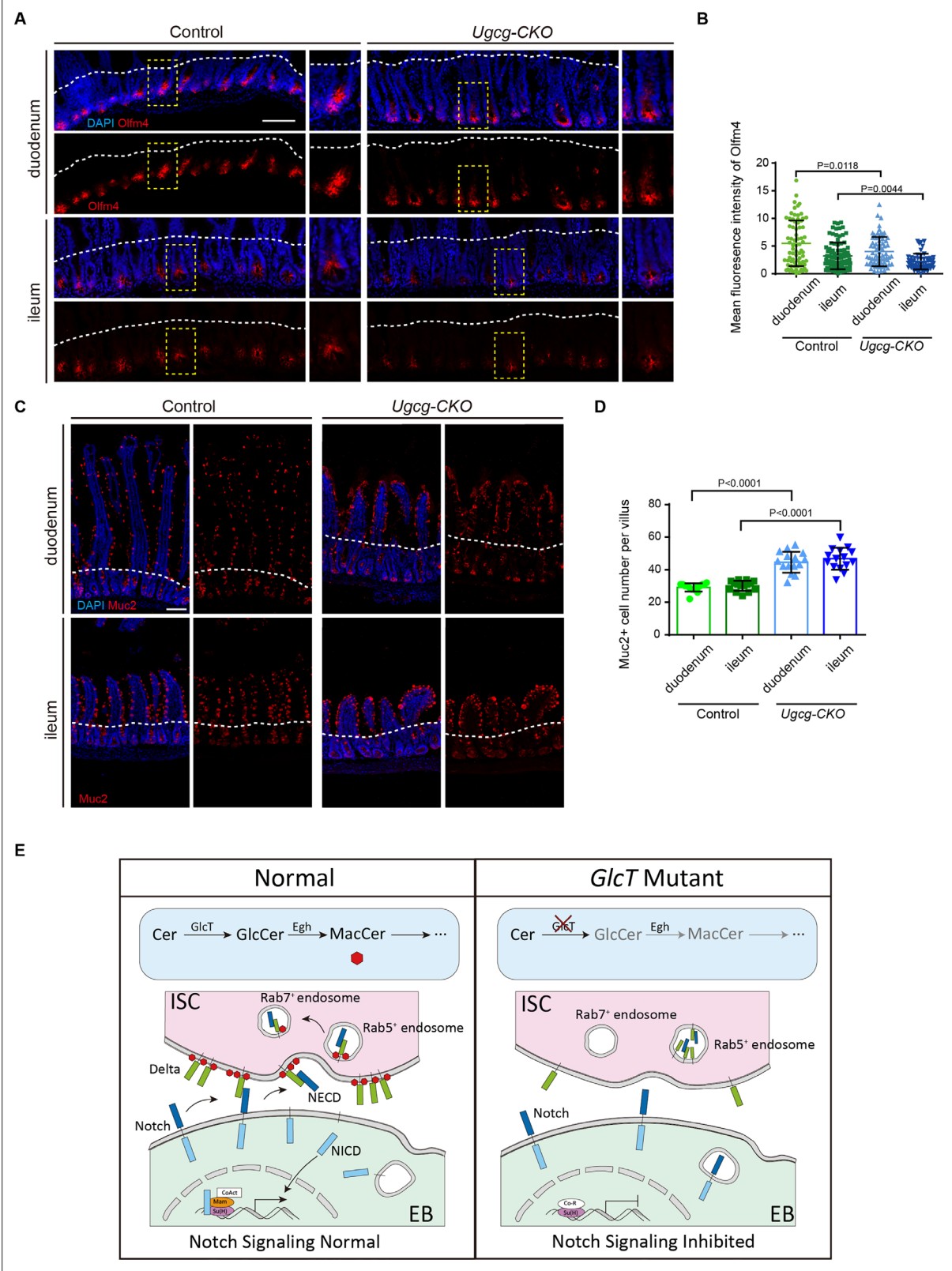

**Figure 7.** Transient loss of *Ugcg* in mouse small intestine causes reduced number of intestinal stem cells (ISCs) and increased number of goblet cells. (**A, B**) Olfm4 staining in the duodenum and ileum of *Villin^CreERT2; Ugcg^flox/flox* mice 48 hours after tamoxifen injection (**A**, n=3) and quantification of Olfm4 fluorescence intensity (**B**). (**C, D**) Muc2 staining of goblet cells in the duodenum and ileum of *Ugcg-CKO* mice (**C**) and quantification of Muc2⁺ goblet

*Figure 7 continued*

cells per villus (**D**, n=3). (**E**) Schematic model illustrating the role of MacCer in regulating the Notch signaling pathway. Error bars represent mean ± SEM, with p-values indicated (two-tailed Student's *t*-test). Scale bars: 100 μm.

The online version of this article includes the following figure supplement(s) for figure 7:

**Figure supplement 1.** The Crypt-Villus morphology in *Ugcg* CKO mice.

endocytosis dysfunctions of Dl caused by the inhibition of the E3 ligase Mib1 or Neur activity (*Hamel et al., 2010*). Therefore, the robust genetic results presented in this study provide definitive evidence for a functional role of MacCer and its downstream derivatives in regulating Notch signaling. Future structural studies focusing on the interactions between GSLs and Dl should offer valuable insights into the role of GSLs in modulating Dl-Notch interactions and membrane trafficking processes.

One intriguing observation from our study is the tissue-specific regulatory role of *GlcT* in Notch signaling. Its loss in ISCs significantly impairs Notch signaling, leading to the formation of EE tumors. However, when *GlcT* is lost in the germline cyst within the developing egg chamber, it only modestly disrupts the endocytic trafficking of Dl, without conspicuous effects on the differentiation and patterning of surrounding follicle cells. In the context of imaginal discs, the loss of *GlcT* appears to have no discernible impact on Notch signaling activity at all. This tissue-specific modulation of Notch signaling by *GlcT* suggests a potential variability in membrane lipid composition across different cell types. Such variability could render certain cells more susceptible to alterations in specific GSLs and their influence on Dl-Notch signaling transduction, while others remain unaffected. As Dl trafficking in *Drosophila* ISCs can be modulated by dietary lipid content, which impacts the duration and level of Notch activation to control the production of EEs in newly enclosed flies (*Obniski et al., 2018*), the sensitivity of ISCs to specific GSL content may serve as a mechanism enabling ISCs to govern the fate of their progeny based on nutrient content.

Taken together, our study identified MacCer/LacCer as a tissue-specific regulator of Dl-Notch signaling by regulating the membrane localization and endocytic trafficking of Dl. As Notch signaling has pleiotropic roles in many developmental processes, diseases, and cancer, the findings should contribute to our understanding of Notch signaling transduction and may help to open up new approaches for developing prevention and treatments against human diseases and cancer.

## Materials and methods
### *Drosophila* stocks and mice

*Drosophila melanogaster* stocks were maintained on standard cornmeal molasses food in a 25°C incubator. Flies for all experiments were maintained at a constant 25°C with 15–25 flies per vial and tossed onto new food every 2–3 days. *Drosophila* stocks used in this study: *hsflp, act-GAL4, UAS-GFP; FRT42B tub-GAL80; hsflp, act-GAL4, UAS-GFP; FRT42D tub-GAL80;* and *hsflp, Tub-GAL80, FRT19A; act-GAL4,UAS-GFP/Cyo* (*Lin et al., 2008*); *hsflp; FRT42D, arm-lacZ/Cyo; esg-GAL4, UAS-GFP* (*Lin et al., 2008*); *Su(H)GBE (NRE)-GAL4;* and *Dl-Gal4* (*Zeng et al., 2010*); *Su(H)GBE-lacZ* (NRE-lacZ, gift from Sarah Bray); *Glc^{EA30}* (generated in this study); *Glc^{E230}* (generated in this study); *GlcT^{Δ8}* (*Satoh et al., 2013*); *Tub-GAL80^{ts}* (*McGuire et al., 2004*); *UAS-GlcT* (generated in this study); *GlcT-IR* #1 (VDRC, id:44912); *GlcT-IR* #2 (VDRC, id:108064); *UAS-CDase* (gift from Tao Wang); *UAS-p35* (BDSC, #6298); *egh^A* (BDSC, #52353); *egh^B* (BDSC, #52354); *egh^7* (BDSC, #77889); *brn^{228}* (BDSC, #7392); *β4GalNAc-TA-IR* (VDRC, id:4867); *α4GT1-IR* (VDRC, id:2608); *UAS-N^{intra}* (*Lin et al., 2008*); *UAS-GFP-Rab5* (BDSC, #43336); *UAS-Rab7-GFP* (BDSC, #42706); *UAS-Rab11-GFP* (BDSC, #8506).

Mice used in this study include floxed *Ugcg* and *Villin^{CreERT2}* (*el Marjou et al., 2004*). Floxed *Ugcg* was generated as previously described (*Jennemann et al., 2012*), but with a modification: we floxed exon 3 (encoding the D1 motif) rather than exons 6–8 (encoding the D3 and (Q/R)XXRW motifs), all of which are critical components of Glucosylceramide synthase active site involved in catalysis and UDP-sugar binding. All the research performance underwent strict ethical review and was approved by the Ethics Committee of the National Institute of Biological Sciences, Beijing. Mice were housed at the animal center in the National Institute of Biological Sciences in a Specific Pathogen Free (SPF) facility. Animal care and use followed the institutional guidelines of the National Institute of Biological Sciences

(NIBS), Beijing (approval ID: NIBS2024M040), and the authors affirm that the study is compliant with all relevant ethical regulations regarding animal research.

## EMS mutagenesis in *Drosophila*

EMS mutagenesis was performed as previously described (*Perdigoto et al., 2011*; *Lee and Luo, 1999*). Briefly, 3–5-day-old, isogenized y w; FRT42B(G13) males were subjected to a 10-hour starvation period before being fed with EMS (ethyl methanesulfonate, 30 mM in sucrose solution applied on filter papers) overnight. Following recovery, these males were mated with *y w; Sco/Cyo* virgin females. In the F1 offspring, *y w; FRT42B */Cyo* males were individually mated with several *y w; Sco/Cyo* virgin females. The *y w; FRT42B */Cyo* males and virgin females in the F2 generation were carefully selected and crossed with each other to establish balanced stocks for further genetic studies. Altogether, around 10,000 stocks harboring lethal mutations were obtained and screened.

## Deficiency mapping

The complete Bloomington Deficiency Kit for chromosome 2R (https://bdsc.indiana.edu/stocks/df/dfkit-info.html) was used for initial mapping, which helped to localize both the EA30 and E230 lethal mutations to a cytogenetic location between 58A3 and 58F1. Subsequent mapping with additional deficiency lines encompassing this region allowed the identification of two small deficiency lines, *Df(2R)Exel7170* and *Df(2R)Excel6078*, which failed to complement both mutations. This refinement narrowed the critical region down to 58B2-58C1. There are 13 protein-coding genes within this region, and genomic sequencing data indicated that both lines carried missense mutations on the *GlcT* gene, as described in the text.

## Genomic sequencing and analysis method

As all mutants exhibited lethality in the adult stage, homozygous larvae were selected using a green balancer, and the extracted genomic DNAs were sequenced at BGI Genomics using a library with an insert size of ≤800 bp. The sequencing platform employed was the HiSeq 2000. For sequencing analysis, sequencing reads from each mutant were aligned to the reference genome separately by Bowtie2 (*Langmead et al., 2019*), then used GATK (*McKenna et al., 2010*) to call variants to get the candidate heterozygous sites, and finally used snpEff (*Cingolani et al., 2012*) to annotate those variants.

## MARCM analysis

To generate MARCM clones (*Lee and Luo, 1999*), flies of the desired genotype were raised and maintained at 25°C until they reached 7 days of age. Subsequently, they were subjected to a 60-minute heat shock in a water bath at 37°C. Following the heat shock, the flies were fed with standard food supplemented with yeast paste, and this feeding regimen was repeated every 2 days before the flies were dissected for further analysis.

## GAL4/GAL80^ts system

The GAL4/UAS and GAL80ts systems were utilized as previously described (*McGuire et al., 2004*; *Brand and Perrimon, 1993*). To inhibit the GAL4 system, the crosses were sustained at 18°C when employing temperature-sensitive GAL4-mediated RNAi or gene overexpression. Subsequently, F1 adult flies with the correct genotype were transitioned to 29°C to activate the GAL4 system, thereby triggering RNAi or gene overexpression.

## Tamoxifen treatment in mice

Cre recombinase activity was induced via intraperitoneal injection of 2 mg tamoxifen (Sigma-Aldrich, T5648), and tissues were collected 48 hours post-injection. Cre-negative littermates served as controls.

## Immunofluorescent staining and imaging

Adult female midguts were dissected in PBS and fixed for 30 minutes at room temperature (RT) in 4% paraformaldehyde. The fixed tissues were then dehydrated in methanol for 5 minutes followed by rehydration in a PBT solution (PBS containing 0.1% Triton X-100). The tissues underwent four washes with 0.1% Triton in 1×PBS (PBST) and were subsequently incubated with PBST and primary

antibodies overnight at 4°C. After another round of washing, the samples were incubated for 2 hours with secondary antibodies in PBST. Finally, the samples were stained with DAPI for visualization of nuclei.

Mouse small intestine was dissected out and gently flushed with cold 1×PBS to remove fecal content and fixed overnight at 4°C in 4% paraformaldehyde (PFA) prepared in 1×PBS. The tissue was then incubated in a 30% sucrose solution at 4°C for 24 hours and embedded in OCT (Tissue-Tek) before frozen. Frozen sections of 15 μm thickness were prepared for further analysis. For immunostaining, tissue sections were permeabilized with 1% Triton X-100 (Sigma-Aldrich) in PBS for 1 hour at RT, followed by blocking with a solution containing 0.2% Triton X-100, 1% bovine serum albumin, and 3% goat serum in PBS for 1 hour at RT before antibody staining. The sections were mounted in Vectashield mounting media with DAPI (Vector Laboratories) for imaging.

Primary antibodies used in this study: mouse anti-Pros (DSHB #MR1A; 1:300); mouse anti-Dl (DSHB Cat#C594.9B; RRID:AB_528194; 1:300); rabbit anti-AstC (gift from Dr. Dick Nassel; 1:300); rabbit anti-Tk (gift from Dr. Jan-Adrianus Veenstra; 1:300); rabbit anti-pH3 (CST Cat# 9701; RRID:AB_331535; 1:500); mouse monoclonal anti-β-galactosidase (DSHB, #40-1a; 1:30); mouse anti-Cut (DSHB, #2B10; 1:20); mouse anti-Rab7 (DSHB, 1:300); rabbit anti-pERK (CST, 1:200); rabbit anti-STAT92E (gift from Dr. Zhaohui Wang, 1:300); rabbit anti-DCP-1 (CST Cat#9578S; 1:300); mouse monoclonal anti-EYA (DSHB, #10H6; 1:300); rabbit monoclonal anti-GFP (Invitrogen Cat#G10362; 1:300). Secondary antibodies used in this study include Alexa Fluor 568- or Cy5-conjugated goat anti-rabbit, anti-mouse IgGs (Molecular Probes, A11034-A11036, A10524; 1:300). For nuclei staining, DAPI (Sigma-Aldrich, 1 μg/ml) was used. The preparations were mounted in 70% glycerol, and the slides were stored at –20°C. Imaging was conducted using a Leica SP8 confocal microscope. All captured images were processed and compiled using Adobe Photoshop and Illustrator.

## LacCer treatment

The 2–3-day-old female flies of the specified genotype were moved to fresh food layered with a filter paper soaked in a 5% sucrose solution containing 20 μM of LacCer (Sigma-Aldrich CAS#4682-48-8), and this transfer was repeated daily before dissection and analysis.

## Antibody uptake assay

The anti-Dl antibody uptake assay was conducted following a previously described protocol (*Couturier and Schweisguth, 2014*). In brief, the *Drosophila* gut was dissected and placed in fresh medium containing anti-Delta. The dissecting dish was then placed in a humidified box and incubated at 25°C for the necessary duration. Subsequently, the medium containing the antibody was removed, and a single wash with Schneider's *Drosophila* medium was carried out. The midgut was then fixed directly in the dissecting dish for 20 minutes under a chemical fume hood by adding 1 ml of fixative. Following this, samples underwent three washes with PBT and were then incubated for 60 minutes with fluorescently labeled anti-mouse secondary antibodies in PBT on a rotating platform. After another three washes with PBT, the samples were mounted in 50% glycerol in PBS for analysis under microscopy.

## Fluorescence intensity statistics

The ImageJ software (downloaded from https://imagej.nih.gov/ij/) was utilized to measure the fluorescence intensity of the captured images. The corrected fluorescence intensity is determined as the average fluorescence intensity of each cell's nuclear region minus the average background fluorescence intensity.

## Statistical analysis

All quantifications were presented in the form of mean ± SEM. GraphPad Prism 5 software (GraphPad Software Inc) was used to calculate p-values using an unpaired Student's *t*-test. The number of intestines used for calculations was labeled on the figures or in the related figure legends.

## Acknowledgements

We thank Drs. Feng Yu and Sanduo Zheng for technical assistance, Drs. Dick Nassel, Jan-Adrianus Veenstra, Zhaohui Wang, and the Developmental Studies Hybridoma Bank (DSHB) for antibodies, Drs. Sarah Bray, Steven Hou, Akiko Satoh, Tao Wang, the Bloomington *Drosophila* Stock Center (BDSC)

and Vienna Drosophila Resource Center (VDRC) for fly stocks, and Dr. Sylvie Robine for the VillinCreERT2 mice. This work was supported by the National Key Research and Development Program of China (2020YFA0803502 and 2017YFA0103602 to RX) from the Chinese Ministry of Science and Technology.

## Additional information

### Funding

| Funder | Grant reference number | Author |
|---|---|---|
| Ministry of Science and Technology of the People's Republic of China | 2020YFA0803502 | Rongwen Xi |
| Ministry of Science and Technology of the People's Republic of China | 2017YFA0103602 | Rongwen Xi |

The funders had no role in study design, data collection and interpretation, or the decision to submit the work for publication.

### Author contributions

Kebei Tang, Conceptualization, Data curation, Formal analysis, Validation, Investigation, Visualization, Methodology, Writing – original draft, Writing – review and editing; Xuewen Li, Conceptualization, Data curation, Formal analysis, Investigation, Visualization, Methodology, Writing – original draft; Jiulong Hu, Conceptualization, Data curation, Formal analysis, Validation, Investigation, Visualization, Methodology, Writing – original draft; Jingyuan Shi, Data curation, Formal analysis, Validation, Investigation, Visualization, Methodology; Yumei Li, Formal analysis, Validation, Investigation; Yansu Chen, Formal analysis, Investigation, Visualization, Methodology; Chang Yin, Formal analysis, Validation; Fengchao Wang, Resources, Supervision, Methodology; Rongwen Xi, Conceptualization, Formal analysis, Supervision, Funding acquisition, Writing – original draft, Writing – review and editing

### Author ORCIDs

Kebei Tang https://orcid.org/0009-0004-3269-4411
Fengchao Wang https://orcid.org/0000-0002-3595-2859
Rongwen Xi https://orcid.org/0000-0001-5543-1236

### Ethics

All the research performance underwent strict ethical review and was approved by the Ethics Committee of the National Institute of Biological Sciences, Beijing. Mice were housed at the animal center in the National Institute of Biological Sciences in a Specific Pathogen Free (SPF) facility. Animal care and use followed the institutional guidelines of the National Institute of Biological Sciences (NIBS), Beijing (approval ID: NIBS2024M040).

Reviewer #1 (Public review): https://doi.org/10.7554/eLife.106184.3.sa1
Reviewer #2 (Public review): https://doi.org/10.7554/eLife.106184.3.sa2
Author response https://doi.org/10.7554/eLife.106184.3.sa3

## Additional files

### Supplementary files

Supplementary file 1. Summary of tumor suppressive gene loci identified in the screen.
MDAR checklist
Source data 1. Raw data of the experiments presented in this article.

### Data availability

All data generated or analyzed during this study are included in the manuscript and supporting files.

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
