## [Editor Report · eLife Assessment]

This **important** study provides **convincing** evidence that glucosylceramide synthase (GlcT), a rate-limiting enzyme for glycosphingolipid (GSL) production, plays a role in the differentiation of intestinal cells. Mutations in GlcT compromise Notch signaling in the *Drosophila* intestinal stem cell lineage, resulting in the formation of enteroendocrine tumors. Further data suggest that a homolog of glucosylceramide synthase also influences Notch signaling in the mammalian intestine. While the outstanding strengths of the initial genetic and downstream pathway analyses are noted, there are minor weaknesses in the data regarding the potential role of this pathway in Delta trafficking. Nevertheless, this study opens the way for future mechanistic studies addressing how specific lipids modulate Notch signaling activity.

---

## [Referee Report · Reviewer #1 (Public review)]

Summary:

From a forward genetic mosaic mutant screen using EMS, the authors identify mutations in glucosylceramide synthase (GlcT), a rate-limiting enzyme for glycosphingolipid (GSL) production, that result in ee tumors. Multiple genetic experiments strongly support the model that the mutant phenotype caused by GlcT loss is due to by failure of conversion of ceramide into glucosylceramide. Further genetic evidence suggests that Notch signaling is comprised in the ISC lineage and may affect endocytosis of Delta. Loss of GlcT does not affect wing development or oogenesis, suggesting tissue-specific roles for GlcT. Finally, an increase in goblet cells in UGCG knockout mice, not previously reported, suggests a conserved role for GlcT in Notch signaling in intestinal cell lineage specification.

Strengths:

Overall, this is a well-written paper with multiple well-designed and executed genetic experiments that support a role for GlcT in Notch signaling in the fly and mammalian intestine. The authors have addressed my concerns from the prior review.

---

## [Referee Report · Reviewer #2 (Public review)]

Summary:

This study genetically identifies two key enzymes involved in the biosynthesis of glycosphingolipids, GlcT and Egh, act as tumor suppressors in the adult fly gut. Detailed genetic analysis indicates that a deficiency in Mactosyl-ceramide (Mac-Cer) is causing tumor formation. Analysis of a Notch transcriptional reporter further indicates that the lack of Mac-Ser is associated with reduced Notch activity in the gut, but not in other tissues.

Addressing how a change in the lipid composition of the membranes might lead to defective Notch receptor activation, the authors studied the endocytic trafficking of Delta and claimed that internalized Delta appeared to accumulate faster into endosomes in the absence of Mac-Cer. Further analysis of Delta steady state accumulation in fixed samples suggested a delay in the endosomal trafficking of Delta from Rab5+ to Rab7+ endosomes, which was interpreted to suggest that the inefficient, or delayed, recycling of Delta might cause a loss in Notch receptor activation.

Finally, the histological analysis of mouse guts following the conditional knock-out of the GlcT gene suggested that Mac-Cer might also be important for proper Notch signaling activity in that context.

Strengths:

The genetic analysis is of high quality. The finding that a Mac-Cer deficiency results in reduced Notch activity in the fly gut is important and fully convincing.

The mouse data, although preliminary, raised the possibility that the role of this specific lipid may be conserved across species.

---

## [Author Response]

The following is the authors’ response to the original reviews.

**Reviewer #1 (Public review):**
Summary:From a forward genetic mosaic mutant screen using EMS, the authors identify mutations in glucosylceramide synthase (GlcT), a rate-limiting enzyme for glycosphingolipid (GSL) production, that result in EE tumors. Multiple genetic experiments strongly support the model that the mutant phenotype caused by GlcT loss is due to by failure of conversion of ceramide into glucosylceramide. Further genetic evidence suggests that Notch signaling is comprised in the ISC lineage and may affect the endocytosis of Delta. Loss of GlcT does not affect wing development or oogenesis, suggesting tissue-specific roles for GlcT. Finally, an increase in goblet cells in UGCG knockout mice, not previously reported, suggests a conserved role for GlcT in Notch signaling in intestinal cell lineage specification.Strengths:Overall, this is a well-written paper with multiple well-designed and executed genetic experiments that support a role for GlcT in Notch signaling in the fly and mammalian intestine. I do, however, have a few comments below.Weaknesses:(1) The authors bring up the intriguing idea that GlcT could be a way to link diet to cell fate choice. Unfortunately, there are no experiments to test this hypothesis.

We indeed attempted to establish an assay to investigate the impact of various diets (such as high-fat, high-sugar, or high-protein diets) on the fate choice of ISCs. Subsequently, we intended to examine the potential involvement of GlcT in this process. However, we observed that the number or percentage of EEs varies significantly among individuals, even among flies with identical phenotypes subjected to the same nutritional regimen. We suspect that the proliferative status of ISCs and the turnover rate of EEs may significantly influence the number of EEs present in the intestinal epithelium, complicating the interpretation of our results. Consequently, we are unable to conduct this experiment at this time. The hypothesis suggesting that GlcT may link diet to cell fate choice remains an avenue for future experimental exploration.

(2) Why do the authors think that UCCG knockout results in goblet cell excess and not in the other secretory cell types?

This is indeed an interesting point. In the mouse intestine, it is well-documented that the knockout of Notch receptors or Delta-like ligands results in a classic phenotype characterized by goblet cell hyperplasia, with little impact on the other secretory cell types. This finding aligns very well with our experimental results, as we noted that the numbers of Paneth cells and enteroendocrine cells appear to be largely normal in UGCG knockout mice. By contrast, increases in other secretory cell types are typically observed under conditions of pharmacological inhibition of the Notch pathway.

(3) The authors should cite other EMS mutagenesis screens done in the fly intestine.

To our knowledge, the EMS screen on 2L chromosome conducted in Allison Bardin’s lab is the only one prior to this work, which leads to two publications (Perdigoto et al., 2011; Gervais, et al., 2019). We have now included citations for both papers in the revised manuscript.

(4) The absence of a phenotype using NRE-Gal4 is not convincing. This is because the delay in its expression could be after the requirement for the affected gene in the process being studied. In other words, sufficient knockdown of GlcT by RNA would not be achieved until after the relevant signaling between the EB and the ISC occurred. Dl-Gal4 is problematic as an ISC driver because Dl is expressed in the EEP.

This is an excellent point, and we agree that the lack of an observable phenotype using NRE-Gal4 could be due to delayed expression, which may result in missing the critical window required for effective GlcT knockdown. Consequently, we cannot rule out the possibility that GlcT also plays a role in early EBs or EEPs. We have revised the manuscript to soften this conclusion and to include this alternative explanation for the experiment.

(5) The difference in Rab5 between control and GlcT-IR was not that significant. Furthermore, any changes could be secondary to increases in proliferation.

We agree that it is possible that the observed increase in proliferation could influence the number of Rab5+ endosomes, and we will temper our conclusions on this aspect accordingly. However, it is important to note that, although the difference in Rab5+ endosomes between the control and GlcT-IR conditions appeared mild, it was statistically significant and reproducible. In our revised experiments, we have not only added statistical data and immunofluorescence images for Rab11 but also unified the approaches used for detecting Rab-associated proteins (in the previous figures, Rab5 was shown using U-Rab5-GFP, whereas Rab7 was detected by direct antibody staining). Based on this unified strategy, we optimized the quantification of Dl-GFP colocalization with early, late, and recycling endosomes, and the results are consistent with our previous observations (see the updated Fig. 5).

**Reviewer #2 (Public review):**
Summary:This study genetically identifies two key enzymes involved in the biosynthesis of glycosphingolipids, GlcT and Egh, which act as tumor suppressors in the adult fly gut. Detailed genetic analysis indicates that a deficiency in Mactosyl-ceramide (Mac-Cer) is causing tumor formation. Analysis of a Notch transcriptional reporter further indicates that the lack of Mac-Ser is associated with reduced Notch activity in the gut, but not in other tissues.Addressing how a change in the lipid composition of the membranes might lead to defective Notch receptor activation, the authors studied the endocytic trafficking of Delta and claimed that internalized Delta appeared to accumulate faster into endosomes in the absence of Mac-Cer. Further analysis of Delta steady-state accumulation in fixed samples suggested a delay in the endosomal trafficking of Delta from Rab5+ to Rab7+ endosomes, which was interpreted to suggest that the inefficient, or delayed, recycling of Delta might cause a loss in Notch receptor activation.Finally, the histological analysis of mouse guts following the conditional knock-out of the GlcT gene suggested that Mac-Cer might also be important for proper Notch signaling activity in that context.Strengths:The genetic analysis is of high quality. The finding that a Mac-Cer deficiency results in reduced Notch activity in the fly gut is important and fully convincing.The mouse data, although preliminary, raised the possibility that the role of this specific lipid may be conserved across species.Weaknesses:This study is not, however, without caveats and several specific conclusions are not fully convincing.First, the conclusion that GlcT is specifically required in Intestinal Stem Cells (ISCs) is not fully convincing for technical reasons: NRE-Gal4 may be less active in GlcT mutant cells, and the knock-down of GlcT using Dl-Gal4ts may not be restricted to ISCs given the perdurance of Gal4 and of its downstream RNAi.

As previously mentioned, we acknowledge that a role for GlcT in early EBs or EEPs cannot be completely ruled out. We have revised our manuscript to present a more cautious conclusion and explicitly described this possibility in the updated version.

Second, the results from the antibody uptake assays are not clear.: (i) the levels of internalized Delta were not quantified in these experiments; (ii) additionally, live guts were incubated with anti-Delta for 3hr. This long period of incubation indicated that the observed results may not necessarily reflect the dynamics of endocytosis of antibody-bound Delta, but might also inform about the distribution of intracellular Delta following the internalization of unbound anti-Delta. It would thus be interesting to examine the level of internalized Delta in experiments with shorter incubation time.

We thank the reviewer for these excellent questions. In our antibody uptake experiments, we noted that Dl reached its peak accumulation after a 3-hour incubation period. We recognize that quantifying internalized Dl would enhance our analysis, and we will include the corresponding statistical graphs in the revised version of the manuscript. In addition, we agree that during the 3-hour incubation, the potential internalization of unbound anti-Dl cannot be ruled out, as it may influence the observed distribution of intracellular Dl. We therefore attempted to supplement our findings with live imaging experiments to investigate the dynamics of Dl/Notch endocytosis in both normal and *GlcT* mutant ISCs. However, we found that the GFP expression level of *Dl-GFP* (either in the knock-in or transgenic line) was too low to be reliably tracked. During the three-hour observation period, the weak GFP signal remained largely unchanged regardless of the *GlcT* mutation status, and the signal resolution under the microscope was insufficient to clearly distinguish membrane-associated from intracellular *Dl*. Therefore, we were unable to obtain a dynamic view of *Dl* trafficking through live imaging. Nevertheless, our *Dl* antibody uptake and endosomal retention analyses collectively support the notion that MacCer influences Notch signaling by regulating *Dl* endocytosis.

Overall, the proposed working model needs to be solidified as important questions remain open, including: is the endo-lysosomal system, i.e. steady-state distribution of endo-lysosomal markers, affected by the Mac-Cer deficiency? Is the trafficking of Notch also affected by the Mac-Cer deficiency? is the rate of Delta endocytosis also affected by the Mac-Cer deficiency? are the levels of cell-surface Delta reduced upon the loss of Mac-Cer?

Regarding the impact on the endo-lysosomal system, this is indeed an important aspect to explore. While we did not conduct experiments specifically designed to evaluate the steady-state distribution of endo-lysosomal markers, our analyses utilizing Rab5-GFP overexpression and Rab7 staining did not indicate any significant differences in endosome distribution in MacCer deficient conditions. Moreover, we still observed high expression of the NRE-LacZ reporter specifically at the boundaries of clones in GlcT mutant cells (Fig. 4A), indicating that GlcT mutant EBs remain responsive to Dl produced by normal ISCs located right at the clone boundary. Therefore, we propose that MacCer deficiency may specifically affect Dl trafficking without impacting Notch trafficking.

In our 3-hour antibody uptake experiments, we observed a notable decrease in cell-surface Dl, which was accompanied by an increase in intracellular accumulation. These findings collectively suggest that Dl may be unstable on the cell surface, leading to its accumulation in early endosomes.

Third, while the mouse results are potentially interesting, they seem to be relatively preliminary, and future studies are needed to test whether the level of Notch receptor activation is reduced in this model.

In the mouse small intestine, Olfm4 is a well-established target gene of the Notch signaling pathway, and its staining provides a reliable indication of Notch pathway activation. While we attempted to evaluate Notch activation using additional markers, such as Hes1 and NICD, we encountered difficulties, as the corresponding antibody reagents did not perform well in our hands. Despite these challenges, we believe that our findings with Olfm4 provide an important start point for further investigation in the future.

**Reviewer #3 (Public review):**
Summary:In this paper, Tang et al report the discovery of a Glycoslyceramide synthase gene, GlcT, which they found in a genetic screen for mutations that generate tumorous growth of stem cells in the gut of Drosophila. The screen was expertly done using a classic mutagenesis/mosaic method. Their initial characterization of the GlcT alleles, which generate endocrine tumors much like mutations in the Notch signaling pathway, is also very nice. Tang et al checked other enzymes in the glycosylceramide pathway and found that the loss of one gene just downstream of GlcT (Egh) gives similar phenotypes to GlcT, whereas three genes further downstream do not replicate the phenotype. Remarkably, dietary supplementation with a predicted GlcT/Egh product, Lactosyl-ceramide, was able to substantially rescue the GlcT mutant phenotype. Based on the phenotypic similarity of the GlcT and Notch phenotypes, the authors show that activated Notch is epistatic to GlcT mutations, suppressing the endocrine tumor phenotype and that GlcT mutant clones have reduced Notch signaling activity. Up to this point, the results are all clear, interesting, and significant. Tang et al then go on to investigate how GlcT mutations might affect Notch signaling, and present results suggesting that GlcT mutation might impair the normal endocytic trafficking of Delta, the Notch ligand. These results (Fig X-XX), unfortunately, are less than convincing; either more conclusive data should be brought to support the Delta trafficking model, or the authors should limit their conclusions regarding how GlcT loss impairs Notch signaling. Given the results shown, it's clear that GlcT affects EE cell differentiation, but whether this is via directly altering Dl/N signaling is not so clear, and other mechanisms could be involved. Overall the paper is an interesting, novel study, but it lacks somewhat in providing mechanistic insight. With conscientious revisions, this could be addressed. We list below specific points that Tang et al should consider as they revise their paper.Strengths:The genetic screen is excellent.The basic characterization of GlcT phenotypes is excellent, as is the downstream pathway analysis.Weaknesses:(1) Lines 147-149, Figure 2E: here, the study would benefit from quantitations of the effects of loss of brn, B4GalNAcTA, and a4GT1, even though they appear negative.

We have incorporated the quantifications for the effects of the loss of brn, B4GalNAcTA, and a4GT1 in the updated Figure 2.

(2) In Figure 3, it would be useful to quantify the effects of LacCer on proliferation. The suppression result is very nice, but only effects on Pros+ cell numbers are shown.

We have now added quantifications of the number of EEs per clone to the updated Figure 3.

(3) In Figure 4A/B we see less NRE-LacZ in GlcT mutant clones. Are the data points in Figure 4B per cell or per clone? Please note. Also, there are clearly a few NRE-LacZ+ cells in the mutant clone. How does this happen if GlcT is required for Dl/N signaling?

In Figure 4B, the data points represent the fluorescence intensity per single cell within each clone. It is true that a few NRE-LacZ+ cells can still be observed within the mutant clone; however, this does not contradict our conclusion. As noted, high expression of the NRE-LacZ reporter was specifically observed around the clone boundaries in MacCer deficient cells (Fig. 4A), indicating that the mutant EBs can normally receive Dl signal from the normal ISCs located at the clone boundary and activate the Notch signaling pathway. Therefore, we believe that, although affecting Dl trafficking, MacCer deficiency does not significantly affect Notch trafficking.

(4) Lines 222-225, Figure 5AB: The authors use the NRE-Gal4ts driver to show that GlcT depletion in EBs has no effect. However, this driver is not activated until well into the process of EB commitment, and RNAi's take several days to work, and so the author's conclusion is "specifically required in ISCs" and not at all in EBs may be erroneous.

As previously mentioned, we acknowledge that a role for GlcT in early EBs or EEPs cannot be completely ruled out. We have revised our manuscript to present a more cautious conclusion and described this possibility in the updated version.

(5) Figure 5C-F: These results relating to Delta endocytosis are not convincing. The data in Fig 5C are not clear and not quantitated, and the data in Figure 5F are so widely scattered that it seems these co-localizations are difficult to measure. The authors should either remove these data, improve them, or soften the conclusions taken from them. Moreover, it is unclear how the experiments tracing Delta internalization (Fig 5C) could actually work. This is because for this method to work, the anti-Dl antibody would have to pass through the visceral muscle before binding Dl on the ISC cell surface. To my knowledge, antibody transcytosis is not a common phenomenon.

We thank the reviewer for these insightful comments and suggestions. In our in vivo experiments, we observed increased co-localization of Rab5 and Dl in *GlcT* mutant ISCs, indicating that Dl trafficking is delayed at the transition to Rab7⁺ late endosomes, a finding that is further supported by our antibody uptake experiments. We acknowledge that the data presented in Fig. 5C are not fully quantified and that the co-localization data in Fig. 5F may appear somewhat scattered; therefore, we have included additional quantification and enhanced the data presentation in the revised manuscript.

Regarding the concern about antibody internalization, we appreciate this point. We currently do not know if the antibody reaches the cell surface of ISCs by passing through the visceral muscle or via other routes. Given that the experiment was conducted with fragmented gut, it is possible that the antibody may penetrate into the tissue through mechanisms independent of transcytosis.

As mentioned earlier, we attempted to supplement our findings with live imaging experiments to investigate the dynamics of Dl/Notch endocytosis in both normal and GlcT mutant ISCs. However, we found that the GFP expression level of *Dl-GFP* (either in the knock-in or transgenic line) was too low to be reliably tracked. During the three-hour observation period, the weak GFP signal remained largely unchanged regardless of the *GlcT* mutation status, and the signal resolution under the microscope was insufficient to clearly distinguish membrane-associated from intracellular *Dl*. Therefore, we were unable to obtain a dynamic view of *Dl* trafficking through live imaging. Nevertheless, our *Dl* antibody uptake and endosomal retention analyses collectively support the notion that MacCer influences Notch signaling by regulating *Dl* endocytosis.

(6) It is unclear whether MacCer regulates Dl-Notch signaling by modifying Dl directly or by influencing the general endocytic recycling pathway. The authors say they observe increased Dl accumulation in Rab5+ early endosomes but not in Rab7+ late endosomes upon GlcT depletion, suggesting that the recycling endosome pathway, which retrieves Dl back to the cell surface, may be impaired by GlcT loss. To test this, the authors could examine whether recycling endosomes (marked by Rab4 and Rab11) are disrupted in GlcT mutants. Rab11 has been shown to be essential for recycling endosome function in fly ISCs.

We agree that assessing the state of recycling endosomes, especially by using markers such as Rab11, would be valuable in determining whether MacCer regulates Dl-Notch signaling by directly modifying Dl or by influencing the broader endocytic recycling pathway. In the newly added experiments, we found that in *GlcT-IR* flies, *Dl* still exhibits partial colocalization with *Rab11*, and the overall expression pattern of *Rab11* is not affected by *GlcT* knockdown (Fig. 5E-F). These observations suggest that MacCer specifically regulates *Dl* trafficking rather than broadly affecting the recycling pathway.

(7) It remains unclear whether Dl undergoes post-translational modification by MacCer in the fly gut. At a minimum, the authors should provide biochemical evidence (e.g., Western blot) to determine whether GlcT depletion alters the protein size of Dl.

While we propose that MacCer may function as a component of lipid rafts, facilitating Dl membrane anchorage and endocytosis, we also acknowledge the possibility that MacCer could serve as a substrate for protein modifications of Dl necessary for its proper function. Conducting biochemical analyses to investigate potential post-translational modifications of Dl by MacCer would indeed provide valuable insights. We have performed Western blot analysis to test whether *GlcT* depletion affects the protein size of Dl. As shown below, we did not detect any apparent changes in the molecular weight of the Dl protein. Therefore, it is unlikely that MacCer regulates post-translational modifications of Dl.

**Author response image 1. sa3fig1:** To investigate whether MacCer modifies Dl by Western blot. (A) Four lanes were loaded: the first two contained 20 μL of membrane extract (lane 1: GlcT-IR, lane 2: control), while the last two contained 10 μL of membrane extract. (B) Full blot images are shown under both long and shortexposure conditions.

(8) It is unfortunate that GlcT doesn't affect Notch signaling in other organs on the fly. This brings into question the Delta trafficking model and the authors should note this. Also, the clonal marker in Figure 6C is not clear.

In the revised working model, we have explicitly described that the events occur in intestinal stem cells. Regarding Figure 6C, we have delineated the clone with a white dashed line to enhance its clarity and visual comprehension.

(9) The authors state that loss of UGCG in the mouse small intestine results in a reduced ISC count. However, in Supplementary Figure C3, Ki67, a marker of ISC proliferation, is significantly increased in UGCG-CKO mice. This contradiction should be clarified. The authors might repeat this experiment using an alternative ISC marker, such as Lgr5.

Previous studies have indicated that dysregulation of the Notch signaling pathway can result in a reduction in the number of ISCs. While we did not perform a direct quantification of ISC numbers in our experiments, our Olfm4 staining—which serves as a reliable marker for ISCs—demonstrates a clear reduction in the number of positive cells in UGCG-CKO mice.

The increased Ki67 signal we observed reflects enhanced proliferation in the transit-amplifying region, and it does not directly indicate an increase in ISC number. Therefore, in UGCG-CKO mice, we observe a decrease in the number of ISCs, while there is an increase in transit-amplifying (TA) cells (progenitor cells). This increase in TA cells is probably a secondary consequence of the loss of barrier function associated with the UGCG knockout.